# Exclusive Feature Learning on Arbitrary Structures via $\ell_{1,2}$-norm

**Deguang Kong**[1]**, Ryohei Fujimaki**[2]**, Ji Liu**[3]**, Feiping Nie**[1]**, Chris Ding**[1]

[1] Dept. of Computer Science, University of Texas Arlington, TX, 76019;
[2] NEC Laboratories America, Cupertino, CA, 95014;
[3] Dept. of Computer Science, University of Rochester, Rochester, NY, 14627
Email: doogkong@gmail.com, rfujimaki@nec-labs.com,
jliu@cs.rochester.edu, feipingnie@gmail.com, chqding@uta.edu

## Abstract

Group LASSO is widely used to enforce the structural sparsity, which achieves the sparsity at the inter-group level. In this paper, we propose a new formulation called "exclusive group LASSO", which brings out sparsity at intra-group level in the context of feature selection. The proposed exclusive group LASSO is applicable on any feature structures, regardless of their overlapping or non-overlapping structures. We provide analysis on the properties of exclusive group LASSO, and propose an effective iteratively re-weighted algorithm to solve the corresponding optimization problem with rigorous convergence analysis. We show applications of exclusive group LASSO for uncorrelated feature selection. Extensive experiments on both synthetic and real-world datasets validate the proposed method.

## 1 Introduction

Structure sparsity induced regularization terms [1, 8] have been widely used recently for feature learning purpose, due to the inherent sparse structures of the real world data. Both theoretical and empirical studies have suggested the powerfulness of structure sparsity for feature learning, e.g., Lasso [24], group LASSO [29], exclusive LASSO [31], fused LASSO [25], and generalized LASSO [22]. To make a compromise between the regularization term and the loss function, the sparse-induced optimization problem is expected to fit the data with better statistical properties. Moreover, the results obtained from sparse learning are easier for interpretation, which give insights for many practical applications, such as gene-expression analysis [9], human activity recognition [14], electronic medical records analysis [30], *etc*.

**Motivation** Of all the above sparse learning methods, group LASSO [29] is known to enforce the sparsity on variables at an inter-group level, where variables from different groups are competing to survive. Our work is motivated from a simple observation: in practice, not only features from different groups are competing to survive (i.e., group LASSO), but also features in a seemingly cohesive group are competing to each other. The winner features in a group are set to large values, while the loser features are set to zeros. Therefore, it leads to sparsity at the intra-group level. In order to make a distinction with standard LASSO and group LASSO, we called it "exclusive group LASSO" regularizer. In "exclusive group LASSO" regularizer, intra-group sparsity is achieved via $\ell_1$ norm, while inter-group non-sparsity is achieved via $\ell_2$ norm. Essentially, standard group LASSO achieves sparsity via $\ell_{2,1}$ norm, while the proposed exclusive group LASSO achieves sparsity via $\ell_{1,2}$ norm. An example of exclusive group LASSO is shown in Fig.(1) via Eq.(2). The significant difference from the standard LASSO is to encourage similar features in different groups to co-exist (Lasso usually allows only one of them surviving). Overall, the exclusive group LASSO regularization encourages intra-group competition but discourages inter-group competition.

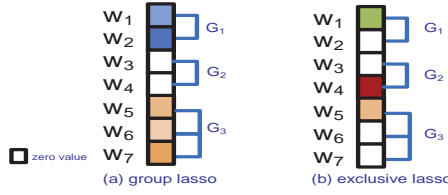

Figure 1: Explanation of differences between group LASSO and exclusive group LASSO. Group setting: $\mathcal{G}_1 = \{1,2\}, \mathcal{G}_2 = \{3,4\}, \mathcal{G}_3 = \{5,6,7\}$. Group LASSO solution of Eq.(3) at $\lambda = 2$ using least square loss is: $\mathbf{w} = [0.0337; 0.0891; 0; 0; -0.2532; 0.043; 0.015]$. exclusive group LASSO solution of Eq.(2) at $\lambda = 10$ is: $\mathbf{w} = [0.0749; 0; 0; -0.0713; -0.1888; 0; 0]$. Clearly, group LASSO introduces sparsity at an *inter-group* level, whereas exclusive LASSO enforces sparsity at an *intra-group* level.

We note that "exclusive LASSO" was first used in [31] for multi-task learning. Our "exclusive group LASSO" work, however, has clear difference from [31]: (1) we give a clear physical intuition of "exclusive group LASSO", which leads to sparsity at an intra-group level (Eq.2), whereas [31] focuses on "Exclusive LASSO" problem in a multi-task setting; (2) we target a general "group" setting which allows arbitrary group structure, which can be easily extended to multi-task/multi-label learning. The main contributions of this paper include: **(1)** we propose a new formulation of "exclusive group LASSO" with clear physical meaning, which allow any arbitrary structure on feature space; **(2)** we propose an effective iteratively re-weighted algorithm to tackle non-smooth "exclusive group LASSO" term with rigorous convergence guarantees. Moreover, an effective algorithm is proposed to handle both non-smooth $\ell_1$ and exclusive group LASSO term (Lemma 4.1); **(3)** The proposed approach is validated via experiments on both synthetic and real data sets, specifically for uncorrelated feature selection problems.

**Notation** Throughout the paper, matrices are written as boldface uppercase, vectors are written as boldface lowercase, and scalars are denoted by lower-case letters $(a, b)$. $n$ is the number of data points, $p$ is the dimension of data, $K$ is the number of class in a dataset. For any vector $\mathbf{w} \in \Re^p$, $\ell_q$ norm of $\mathbf{w}$ is $\|\mathbf{w}\|_q = \left( \sum_{i=1}^{p} |w_j|^q \right)^{\frac{1}{q}}$ for $q \in (0, \infty)$. A group of variables is a subset $g \subset \{1, 2, \cdots, p\}$. Thus, the set of possible groups is the power set of $\{1, 2, \cdots, p\}$: $\mathcal{P}(\{1, 2, \cdots, p\})$. $\mathcal{G}_g \in \mathcal{P}(\{1, 2, \cdots, p\})$ denotes a set of group $g$, which is known in advance depending on applications. If two groups have one or more overlapped variables, we say that they are overlapped. For any group variable $\mathbf{w}_{\mathcal{G}_g} \in \Re^p$, only the entries in the group $g$ are preserved which are the same as those in $\mathbf{w}$, while the other entries are set to zeros. For example, if $\mathcal{G}_g = \{1, 2, 4\}$, $\mathbf{w}_{\mathcal{G}_g} = [w_1, w_2, 0, w_4, 0, \cdots, 0]$, then $\|\mathbf{w}_{\mathcal{G}_g}\|_2 = \sqrt{w_1^2 + w_2^2 + w_4^2}$. Let $supp(\mathbf{w}) \subset \{1, 2, \cdots, p\}$ be a set which $w_i \neq 0$, and $zero(\mathbf{w}) \subset \{1, 2, \cdots, p\}$ be a set which $w_i = 0$. Clearly, $zero(\mathbf{w}) = \{1, 2, \cdots, p\} \setminus supp(\mathbf{w})$. Let $\bigtriangledown f(\mathbf{w})$ be gradient of $f$ at $\mathbf{w} \in \Re^p$, for any differentiable function $f$: $\Re^p \to \Re$.

## 2 Exclusive group LASSO

Let $\mathcal{G}$ be a group set, the exclusive group LASSO penalty is defined as:

$$\forall \mathbf{w} \in \Re^p, \ \ \Omega_{Eg}^{\mathcal{G}}(\mathbf{w}) = \sum_{g \in \mathcal{G}} \|\mathbf{w}_{\mathcal{G}_g}\|_1^2. \tag{1}$$

When the groups of $g$ form different partitions of the set of variables, $\Omega_{Eg}^{\mathcal{G}}$ is a $\ell_1/\ell_2$ norm penalty. A $\ell_2$ norm is enforced on different groups, while in each group, $\ell_1$ norm is used to make a sum over each intra-group variable. Minimizing such a convex risk function often leads to a solution that some entries in a group are zeros. For example, for a group $\mathcal{G}_g = \{1, 2, 4\}$, there exists a solution $\mathbf{w}$, such that $w_1 = 0, w_2 \neq 0, w_4 \neq 0$. A concrete example is shown in Fig.1, in which we solve:

$$\min_{\mathbf{w} \in \Re^p} J_1(\mathbf{w}), \ \ J_1(\mathbf{w}) = f(\mathbf{w}) + \lambda \Omega_{Eg}^{\mathcal{G}}(\mathbf{w}). \tag{2}$$

using least square loss function $f(\mathbf{w}) = \|\mathbf{y} - \mathbf{X}^T \mathbf{w}\|_2^2$. As compared to standard group LASSO [29] solution of Eq.(3),

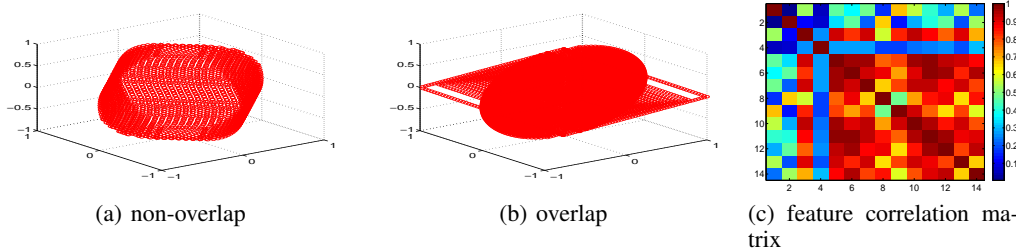

| (a) non-overlap | (b) overlap | (c) feature correlation matrix |

Figure 2: (a-b): Geometric shape of $\Omega(\mathbf{w}) \le 1$ in $\mathbf{R}^3$. (a) non-overlap exclusive group LASSO: $\Omega(\mathbf{w}) = (|\mathbf{w}_1| + |\mathbf{w}_2|)^2 + (|\mathbf{w}_3|)^2$; (b) overlap exclusive group LASSO: $\Omega(\mathbf{w}) = (|\mathbf{w}_1| + |\mathbf{w}_2|)^2 + (|\mathbf{w}_2| + |\mathbf{w}_3|)^2$; (c) feature correlation matrix $\mathbf{R}$ on dataset House (506 data points, 14 variables). $R_{ij}$ indicates the feature correlation between feature $i$ and $j$. *Red* colors indicate large values, while *blue* colors indicate small values.

$$f(\mathbf{w}) + \lambda \sum_g \|\mathbf{w}_{\mathcal{G}_g}\|_2. \tag{3}$$

We observe that group LASSO introduces sparsity at an *inter-group level*, whereas exclusive LASSO enforces sparsity at an *intra-group level*.

**Analysis of exclusive group LASSO** For each group $g$, feature index $u \in supp(g)$ will be non-zero. Let $\mathbf{v}_g \in \Re^p$ be a variable which preserves the values of non-zero index for group $g$. Consider all groups, for optimization goal $\mathbf{w}$, we have $supp(\mathbf{w}) = \underset{g}{\cup} supp(\mathbf{v}_g)$. **(1)** For non-overlapping case, different groups form a partition of feature set $\{1, 2, \cdots, p\}$, and there exists a unique decomposition of $\mathbf{w} = \sum_g \mathbf{v}_g$. Since there is not any common elements for any two different groups $\mathcal{G}_u$ and $\mathcal{G}_v$, i.e., $supp(\mathbf{w}_{\mathcal{G}_u}) \cap supp(\mathbf{w}_{\mathcal{G}_v}) = \phi$. thus it is easy to see: $\mathbf{v}_g = \mathbf{w}_{\mathcal{G}_g}, \forall g \in \mathcal{G}$. **(2)** However, for overlapping groups, there could be element sets $I \subset (\mathcal{G}_u \cap \mathcal{G}_v)$, and therefore, different groups $\mathcal{G}_u$ and $\mathcal{G}_v$ may have opposite effects to optimize the features in set $I$. For feature $i \in I$, it is prone to give different values if optimized separately, i.e., $(\mathbf{w}_{\mathcal{G}_u})_i \ne (\mathbf{w}_{\mathcal{G}_v})_i$. For example, $\mathcal{G}_u = [1, 2], \mathcal{G}_v = [2, 3]$, whereas group $u$ may require $w_2 = 0$ and group $v$ may require $w_2 \ne 0$. Thus, there will be many possible combinations of feature values, and it leads to: $\Omega_{Eg}^{\mathcal{G}} = \underset{\sum_g \mathbf{v}_g = \mathbf{w}}{\inf} \sum_g \|\mathbf{v}_g\|_1^2$. Further, if some groups are overlapped, the final zeros sets will be a subset of unions of all different groups. $zero(\mathbf{w}) \subset \underset{g}{\cap} zero(\mathbf{v}_g)$.

**Illustration of Geometric shape of exclusive LASSO** Figure 2 shows the geometric shape for both norms in $\mathbf{R}^3$ with different group settings, where in (a): $\mathcal{G}_1 = [1, 2], \mathcal{G}_2 = [3]$; and in (b): $\mathcal{G}_1 = [1, 2], \mathcal{G}_2 = [2, 3]$. For the non-overlapping case, variables $w_1, w_2$ usually can not be zero simultaneously. In contrast, for the overlapping case, variable $w_2$ cannot be zero unless both groups $\mathcal{G}_1$ and $\mathcal{G}_2$ require $\mathbf{w}_2 = 0$.

**Properties of exclusive LASSO** The regularization term of Eq.(1) is a convex formulation. If $\cup_{g \in \mathcal{G}} = \{1, 2, \cdots, p\}$, then $\Omega_E^{\mathcal{G}} := \sqrt{\Omega_{Eg}^{\mathcal{G}}}$ is a norm. See Appendix for proofs.

## 3 An effective algorithm for solving $\Omega_{Eg}^{\mathcal{G}}$ regularizer

The challenge of solving Eq. (1) is to tackle the exclusive group LASSO term, where $f(\mathbf{w})$ can be any convex loss function w.r.t $\mathbf{w}$. It is generally felt that exclusive group LASSO term is much more difficult to solve than the standard LASSO term (shrinkage thresholding). Existing algorithm can formulate it as a quadratic programming problem [19], which can be solved by interior point method or active set method. However, the computational cost is expensive, which limits its use in practice. Recently, a primal-dual algorithm [27] is proposed to solve the similar problem, which casts the non-smooth problem into a min-max problem. However, the algorithm is a gradient descent type method and converges slowly. Moreover, the algorithm is designed for multi-task learning problem, and cannot be applied directly for exclusive group LASSO problem with arbitrary structures.

In the following, we *first* derive a very efficient yet simple algorithm. Moreover, the proposed algorithm is a generic algorithm, which allows arbitrary structure on feature space, irrespective of specific feature structures [10], e.g., linear structure [28], tree structure [15], graph structure [7], etc.

Theoretical analysis guarantees the convergence of algorithm. Moreover, the algorithm is easy to implement and ready to use in practice.

**Key idea** The idea of the proposed algorithm is to find an auxiliary function for Eq.(1) which can be easily solved. Then the updating rules for $\mathbf{w}$ is derived. Finally, we prove the solution is exactly the optimal solution we are seeking for the original problem. Since it is a convex problem, the optimal solution is the global optimal solution.

**Procedure** Instead of directly optimizing Eq. (1), we propose to optimize the following objective (the reasons will be seen immediately below), i.e.,

$$J_2(\mathbf{w}) = f(\mathbf{w}) + \lambda \mathbf{w}^T \mathbf{F} \mathbf{w}, \tag{4}$$

where $\mathbf{F} \in \Re^{p \times p}$ is a diagonal matrices which encodes the exclusive group information, and its diagonal element is given by[1]

$$\mathbf{F}_{ii} = \left( \sum_g \frac{(\mathtt{I}_{\mathcal{G}_g})_i \|\mathbf{w}_{\mathcal{G}_g}\|_1}{|\mathbf{w}_i|} \right). \tag{5}$$

Let $\mathtt{I}_{\mathcal{G}_g} \in \{0,1\}^{p \times 1}$ be group index indicator for group $g \in \mathcal{G}$. For example, group $\mathcal{G}_1$ is $\{1,2\}$, then $\mathtt{I}_{\mathcal{G}_1} = [1,1,0,\cdots,0]$. Thus the group variable $\mathbf{w}_{\mathcal{G}_g}$ can be explicitly expressed as $\mathbf{w}_{\mathcal{G}_g} = \mathrm{diag}(\mathtt{I}_{\mathcal{G}_g}) \times \mathbf{w}$.

Note computation of $\mathbf{F}$ depends on $\mathbf{w}$, thus minimization of $\mathbf{w}$ depends on both $\mathbf{F}$. In the following, we propose an efficient *iteratively re-weighted* algorithm to find out the optimal global solution for $\mathbf{w}$, where in each iteration, $\mathbf{w}$ is updated along the gradient descent direction. This process is iterated until the algorithm converges. Taking the derivative of Eq.(4) w.r.t $\mathbf{w}$ and set $\frac{\partial J_2}{\partial \mathbf{w}} = 0$. We have

$$\nabla_{\mathbf{w}} f(\mathbf{w}) + 2\lambda \mathbf{F} \mathbf{w} = 0. \tag{6}$$

Then the complete algorithm is:

(1) Updating $\mathbf{w}^t$ via Eq.(6);

(2) Updating $\mathbf{F}^t$ via Eq.(5).

The above two steps are iterated until the algorithm converges. We can prove the obtained optimal solution is exactly the global optimal solution for Eq.(1).

### 3.1 Convergence Analysis

In the following, we prove the convergence of algorithm.

**Theorem 3.1.** *Under the updating rule of Eq. (6), $J_1(\mathbf{w}^{t+1}) - J_1(\mathbf{w}^t) \leq 0$.*

The proof is provided in Appendix.

**Discussion** We note reweighted strategy [26] was also used in solving problems like zero-norm of the parameters of linear models. However, it cannot be directly used to solve "exclusive group LASSO" problem proposed in this paper, and cannot handle arbitrary structures on feature space.

## 4 Uncorrelated feature learning via exclusive group LASSO

**Motivation** It is known that in Lasso-type (including elastic net) [24, 32] variable selection, variable correlations are not taken into account. Therefore, some strongly correlated variables tend to be in or out of the model together. However, in practice, feature variables are often correlated. See an example shown on housing dataset [4] with 506 samples and 14 attributes. Although there are only 14 attributes, feature 5 is highly correlated with feature 6, 7, 11, 12, etc. Moveover, the strongly correlated variables may share similar properties, with overlapped or redundant information. Especially

Table 1: Characteristics of datasets

| Dataset | # data | #dimension | #domain |
|---------|--------|------------|---------|
| isolet | 1560 | 617 | UCI |
| ionosphere | 351 | 34 | UCI |
| mnist(0,1) | 3125 | 784 | image |
| Leuml | 72 | 3571 | biology |

when the number of selected variables are limited, more discriminant information with minimum correlations are desirable for prediction or classification purpose. Therefore, it is natural to eliminate the correlations in the feature learning process.

**Formulation** The above observations motivate our work of uncorrelated feature learning via exclusive group LASSO. We consider the variable selection problem based on the LASSO-type optimization, where we can make the selected variables uncorrelated as much as possible. To be exact, we propose to optimize the following objective:

$$\min_{\mathbf{w} \in \Re^p} f(\mathbf{w}) + \alpha \|\mathbf{w}\|_1 + \beta \sum_g \|\mathbf{w}_{\mathcal{G}_g}\|_1^2, \tag{7}$$

where $f(\mathbf{w})$ is loss function involving class predictor $\mathbf{y} \in \Re^n$ and data matrix $\mathbf{X} = [\mathbf{x}_1, \mathbf{x}_2, \cdots, \mathbf{x}_n] \in \Re^{p \times n}$, $\|\mathbf{w}_{\mathcal{G}_g}\|_1^2$ is the exclusive group LASSO term involving feature correlation information $\alpha$ and $\beta$ are tuning parameters, which can make balances between plain LASSO term and the exclusive group LASSO term.

The core part of Eq.(7) is to use exclusive group LASSO regularizer to eliminate the correlated features, which cannot be done by plain LASSO. Let the feature correlation matrix be $\mathbf{R} = (R_{st}) \in \Re^{p \times p}$, clearly, $\mathbf{R} = \mathbf{R}^T$, $R_{st}$ represents the correlations between features $s$ and $t$, i.e.,

$$R_{st} = \frac{|\sum_i X_{si} X_{ti}|}{\sqrt{\sum_i X_{si}^2} \sqrt{\sum_i X_{ti}^2}}, \quad R_{st} > \theta \tag{8}$$

To let the selected features uncorrelated as much as possible, for any two features $s, t$, if their correlations $R_{st} > \theta$, then we put them in an exclusive group. Therefore, only one feature can survive. For example, on the example shown in Fig.2(c), if we use $\theta = 0.93$ as a threshold, we will generate the following exclusive group LASSO term:

$$\sum_g \|\mathbf{w}_{\mathcal{G}_g}\|_1^2 \quad = (|w_3| + |w_{10}|)^2 + (|w_5| + |w_6|)^2 + (|w_5| + |w_7|)^2 + (|w_5| + |w_{11}|)^2 + (|w_6| + |w_{11}|)^2$$

$$+ (|w_6| + |w_{12}|)^2 + (|w_6| + |w_{14}|)^2 + (|w_7| + |w_{11}|)^2. \tag{9}$$

**Algorithm** Solving Eq.(7) is to solve a convex optimization problem, because all the three involved terms are convex. This also indicates that there exists a unique global solution. Eq.(7) can be efficiently solved via accelerated proximal gradient (FISTA) method [17, 2], irrespective of what kind of loss function used in minimization of empirical risk. Thus solving Eq.(7) is transformed into solving:

$$\min_{\mathbf{w} \in \Re^p} \frac{1}{2} \|\mathbf{w} - \mathbf{a}\|_2^2 + \alpha \|\mathbf{w}\|_1 + \beta \sum_g \|\mathbf{w}_{\mathcal{G}_g}\|_1^2, \tag{10}$$

where $\mathbf{a} = \mathbf{w}^t - \frac{1}{L_t} \nabla f(\mathbf{w}^t)$ which involves the current $\mathbf{w}^t$ value and step size $L_t$. The challenge of solving Eq.(10) is that, it involves two non-smooth terms. Fortunately, we have the following lemma to establish the relations between the optimal solution of Eq.(10) to Eq.(11), the solution of which has been well discussed in §3.

$$\min_{\mathbf{w} \in \Re^p} \frac{1}{2} \|\mathbf{w} - \mathbf{u}\|_2^2 + \beta \sum_g \|\mathbf{w}_{\mathcal{G}_g}\|_1^2. \tag{11}$$

**Lemma 4.1.** *The optimal solution to Eq.(10) is the optimal solution to Eq.(11), where*

$$\mathbf{u} = \arg \min_{\mathbf{x}} \frac{1}{2} \|\mathbf{x} - \mathbf{a}\|_2^2 + \alpha \|\mathbf{x}\|_1 = sgn(\mathbf{a})(\mathbf{a} - \alpha)_+, \tag{12}$$

*and $sgn(.)$, $SGN(.)$ are the operators defined in the component fashion: if $v > 0$, $sgn(v) = 1$, $SGN(v) = \{1\}$; else if $v = 0$, $sgn(v) = 0$, $SGN(v) = [-1, 1]$; else if $v < 0$, $sgn(v) = -1$, $SGN(v) = \{-1\}$.*

The proof is provided in Appendix.

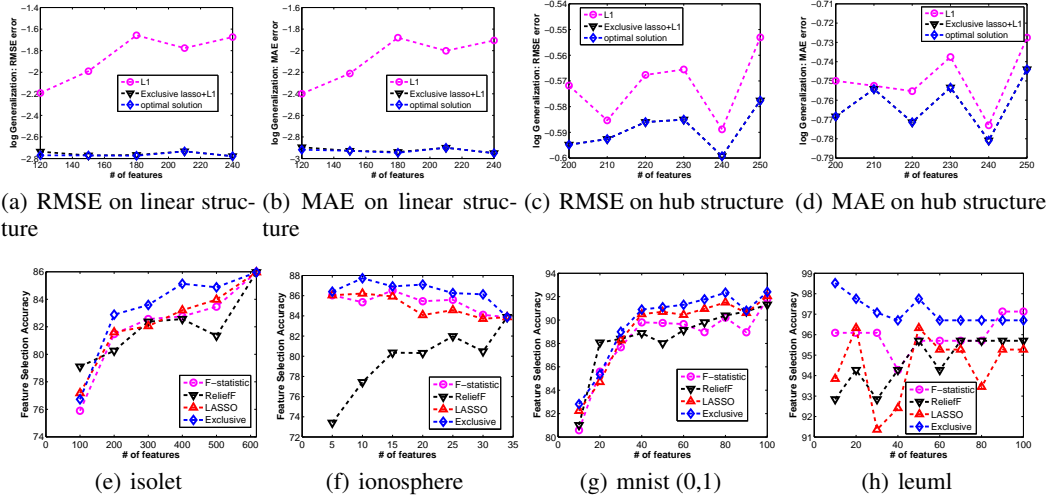

(a) RMSE on linear struc- (b) MAE on linear struc- (c) RMSE on hub structure (d) MAE on hub structure
ture                        ture

(e) isolet          (f) ionosphere          (g) mnist (0,1)          (h) leuml

Figure 3: (a-d): Feature selection results on synthetic dataset using (a, b) linear structure; (c, d) hub structure. Evaluation metrics: RMSE, MAE. x-axis: number of selected features. y-axis: RMSE or MAE error in log scale. (g-j): **Classification accuracy** using SVM (linear kernel) with different number of selected features on four datasets. Compared methods: Exclusive LASSO of Eq.(7), LASSO, ReliefF [21], F-statistics [3]. x-axis: number of selected features; y-axis: **classification accuracy**.

# 5 Experiment Results

To validate the effectiveness of our method, we first conduct experiment using Eq.(7) on two synthetic datasets, and then show experiments on real-world datasets.

## 5.1 Synthetic datasets

**(1) Linear-correlated features**. Let data $\mathbf{X}^1 = [\mathbf{x}_1^1, \mathbf{x}_2^1, \cdots, \mathbf{x}_n^1] \in \Re^{p \times n}$, $\mathbf{X}^2 = [\mathbf{x}_1^2, \mathbf{x}_2^2, \cdots, \mathbf{x}_n^2] \in \Re^{p \times n}$, where each data $\mathbf{x}_i^1 \sim \mathcal{N}[\mathbf{0}_{p \times 1}, \mathbf{I}_{p \times p}]$, $\mathbf{x}_i^2 \sim \mathcal{N}[\mathbf{0}_{p \times 1}, \mathbf{I}_{p \times p}]$, $\mathbf{I}$ is identity matrix. We generate a group of $p$-features, which is a linear combination of features in $\mathbf{X}_1$ and $\mathbf{X}_2$, i.e., $\mathbf{X}_3 = 0.5(\mathbf{X}_1 + \mathbf{X}_2) + \epsilon$, $\epsilon \sim \mathcal{N}(-0.1\mathbf{e}, 0.1\mathbf{I}_{p \times p})$. Construct data matrix $\mathbf{X} = [\mathbf{X}_1; \mathbf{X}_2; \mathbf{X}_3]$, clearly, $\mathbf{X} \in \Re^{3p \times n}$. Features in dimension $[2p + 1, 3p]$ are highly correlated with features in dimension $[1, p]$ and $[p + 1, 2p]$. Let $\mathbf{w}^1 \in \Re^p$, where each $\mathbf{w}_i^1 \sim \text{Uniform}(-0.5, 0.5)$, and $\mathbf{w}^2 \in \Re^p$, where each $\mathbf{w}_i^2 \sim \text{Uniform}(-0.5, 0.5)$. Let $\tilde{\mathbf{w}} = [\mathbf{w}^1; \mathbf{w}^2; \mathbf{0}_{\mathbf{p} \times \mathbf{1}}]$. We generate predicator $\mathbf{y} \in \Re^n$, and $\mathbf{y} = \tilde{\mathbf{w}}^T \mathbf{X} + \epsilon_y$, where $(\epsilon_y)_i \sim \mathcal{N}(0, 0.1)$.

We solve Eq.(7) using current $\mathbf{y}$ and $\mathbf{X}$ with least square loss. The group settings are: $(i, p+i, 2p+i)$, for $1 \le i \le p$. We compare the computed $\mathbf{w}^*$ against ground truth solution $\tilde{\mathbf{w}}$ and plain LASSO solution (i.e., $\beta = 0$ in Eq.7). We use the root mean square error (RMSE) and mean absolute error (MAE) error to evaluate the differences of values predicted by a model and the values actually observed. We generate $n = 1000$ data, with $p = [120, 140, \cdots, 220, 240]$ and do 5-fold cross validation. Generalization error of RMSE and MAE are shown in Figures 3(a) and 3(b). Clearly, our approach outperforms standard LASSO solution and exactly recovers the true features.

**(2) Correlated features on Hub structure**. Let data $\mathbf{X} = [\mathbf{X}^1; \mathbf{X}^2; \cdots, \mathbf{X}^B] \in \Re^{q \times n}$, where each block $\mathbf{X}^b = [X_{1:}^b; X_{2:}^b; \cdots; X_{p:}^b] \in \Re^{p \times n}$, $1 \le b \le B$, $q = p \times B$. In each block, for each data point $1 \le i \le n$, $X_{1i}^b = \frac{1}{B} \sum_{2 \le j \le p} X_{ji}^b + \frac{1}{B} z_i + \epsilon_i^b$, where $X_{ji}^b \sim \mathcal{N}(0, 1)$, $z_i \sim \mathcal{N}(0, 1)$ and $\epsilon_i^b \sim \text{Uniform}(-0.1, 0.1)$. Let $\mathbf{w}^1, \mathbf{w}^2, \cdots, \mathbf{w}^B \in \Re^p$, where $\mathbf{w}^b = [w_1^b \quad \mathbf{0}]^T$, where $w_1^b \sim \text{Uniform}(-0.5, 0.5)$. Let $\tilde{\mathbf{w}} = [\mathbf{w}^1; \mathbf{w}^2; \cdots; \mathbf{w}^B]$, we generate predicator $\mathbf{y} \in \Re^n$, and $\mathbf{y} = \tilde{\mathbf{w}}^T \mathbf{X} + \epsilon_y$, where $(\epsilon_y)_i \sim \mathcal{N}(0, 0.1)$.

The group settings are: $((b - 1) \times p + 1, \cdots, b \times p)$, for $1 \le b \le B$. We generate $n = 1000$ data, $B = 10$, with varied $p = [20, 21, \cdots, 24, 25]$ and do 5-fold cross validation. Generalization error of RMSE and MAE are shown in Figs.3(c),3(d). Clearly, our approach outperforms standard LASSO solution, and recovers the exact features.

## 5.2 Real-world datasets

To validate the effectiveness of proposed method, we perform feature selection via proposed uncorrelated feature learning framework of Eq.(7) on 4 datasets (shown in Table.1), including 2 UCI datasets: isolet [6], ionosphere [5], 1 image dataset: mnist with only figure "0" and "1" [16], and 1 biology dataset: Leuml [13].

We perform classification tasks on these different datasets. The compared methods include: proposed method of Eq.(7) (shown as Exclusive), plain LASSO, ReliefF [21], F-statistics [3]. We use logistic regression as the loss function in our method and plain LASSO method. In our method, parameter $\alpha$, $\beta$ are tuned to select different numbers of features. Exclusive LASSO groups are set according to feature correlations (i.e., threshold $\theta$ is set to 0.90 in Eq.8). After the specific number of features are selected, we feed them into support vector machine (SVM) with linear kernel, and classification results with different number of selected features are shown in Fig.(3).

A first glance at the experimental results indicates the better performance of our method as compared to plain LASSO. Moreover, our method is also generally better than the other two popularly used feature selection methods, such as ReliefF and F-statistics. The experiment result also further confirms our intuition: elimination of correlated features is really helpful for feature selection and thus improves the classification performance. Because $\ell_{1,\infty}$ [20], $\ell_{2,1}$ [12, 18], or non-convex feature learning via $\ell_{p,\infty}$ [11]$(0 < p < 1)$ are designed for multi-task or multi-label feature learning, thus we do not compare against these methods.

Further, we list the mean and variance of classification accuracy of different algorithms in the following table, using 50% of all the features. Compared methods include (1) Lasso (L1); (2) Plain exclusive group LASSO ($\alpha = 0$ in Eq. (7)); (3) Exclusive group LASSO ($\alpha > 0$ in Eq. (7)).

| dataset | # of features | LASSO | plain exclusive | exclusive group LASSO |
|---|---|---|---|---|
| isolet | 308 | $81.75 \pm 0.49$ | $82.05 \pm 0.50$ | $83.24 \pm 0.23$ |
| ionosphere | 17 | $85.10 \pm 0.27$ | $85.21 \pm 0.31$ | $87.28 \pm 0.42$ |
| mnist(0,1) | 392 | $92.35 \pm 0.13$ | $93.07 \pm 0.20$ | $94.51 \pm 0.19$ |
| leuml | 1785 | $95.10 \pm 0.31$ | $95.67 \pm 0.24$ | $97.70 \pm 0.27$ |

The above experiment results indicate that the advantage of our method (exclusive group LASSO) over plain LASSO comes from the exclusive LASSO term. The experiment results also suggest that the plain exclusive LASSO performs very similar to LASSO. However, the exclusive group LASSO ($\alpha > 0$ in Eq.7) performs definitely better than both standard LASSO and plain exclusive LASSO (1%-4% performance improvement). The exclusive LASSO regularizer eliminates the correlated and redundant features.

We show the running time of plain exclusive LASSO and exclusive group LASSO ($\alpha > 0$ in Eq.7) in the following table. We run different algorithms on a Intel i5-3317 CPU, 1.70GHz, 8GRAM desktop.

| dataset | plain exclusive (running time: sec) | exclusive group LASSO (running time: sec) |
|---|---|---|
| isolet | 47.24 | 51.93 |
| ionosphere | 22.75 | 24.18 |
| mnist(0,1) | 123.45 | 126.51 |
| leuml | 142.19 | 144.08 |

The above experiment results indicate that the computational cost of exclusive group LASSO is slightly higher than that of plain exclusive LASSO. The reason is that, the solution to exclusive group LASSO is given by simple thresholding on the plain exclusive LASSO result. This further confirms our theoretical analysis results shown in Lemma 4.1.

## 6 Conclusion

In this paper, we propose a new formulation called "exclusive group LASSO" to enforce the sparsity for features at an intra-group level. We investigate its properties and propose an effective algorithm with rigorous convergence analysis. We show applications for uncorrelated feature selection, which indicate the good performance of proposed method. Our work can be easily extended for multi-task or multi-label learning.

**Acknowledgement** The majority of the work was done during the internship of the first author at NEC Laboratories America, Cupertino, CA.

## Appendix

**Proof of a valid norm of $\Omega_E^{\mathcal{G}}$:** Note if $\Omega_E^{\mathcal{G}}(\mathbf{w}) = 0$, then $\mathbf{w} = \mathbf{0}$. For any scalar $a$, $\Omega_E^{\mathcal{G}}(a\mathbf{w}) = |a|\Omega_E^{\mathcal{G}}(\mathbf{w})$. This proves absolute homogeneity and zero property hold. Next we consider triangle inequality. Consider $\mathbf{w}, \tilde{\mathbf{w}} \in \Re^p$. Let $\mathbf{v}_g$ and $\tilde{\mathbf{v}}_g$ be optimal decomposition of $\mathbf{w}, \tilde{\mathbf{w}}$ such that $\Omega_E^{\mathcal{G}}(\mathbf{w}) = \sqrt{\sum_g \|\mathbf{v}_g\|_1^2}$, and $\Omega_E^{\mathcal{G}}(\tilde{\mathbf{w}}) = \sqrt{\sum_g \|\tilde{\mathbf{v}}_g\|_1^2}$. Since $\mathbf{v}_g + \tilde{\mathbf{v}}_g$ is a decomposition of $\mathbf{w} + \tilde{\mathbf{w}}$, thus we have: [1] $\Omega_E^{\mathcal{G}}(\mathbf{w} + \tilde{\mathbf{w}}) \leq \sqrt{\sum_g \|\mathbf{v}_g + \tilde{\mathbf{v}}_g\|_1^2} \leq \sqrt{\sum_g \|\mathbf{v}_g\|_1^2} + \sqrt{\sum_g \|\tilde{\mathbf{v}}_g\|_1^2} = \Omega_E^{\mathcal{G}}(\mathbf{w}) + \Omega_E^{\mathcal{G}}(\tilde{\mathbf{w}})$. □

To prove Theorem 3.1, we need two lemmas.

**Lemma 6.1.** *Under the updating rule of Eq.(6), $J_2(\mathbf{w}^{t+1}) < J_2(\mathbf{w}^t)$.*

**Lemma 6.2.** *Under the updating rule of Eq.(6),*

$$\left( J_1(\mathbf{w}^{t+1}) - J_1(\mathbf{w}^t) \right) \leq \left( J_2(\mathbf{w}^{t+1}) - J_2(\mathbf{w}^t) \right). \tag{13}$$

**Proof of Theorem 3.1** From Lemma 6.1 and Lemma 6.2, it is easy to see $\left( J_1(\mathbf{w}^{t+1}) - J_1(\mathbf{w}^t) \right) \leq 0$. This completes the proof. □

**Proof of Lemma 6.1** Eq.(4) is a convex function, and optimal solution of Eq.(6) is obtained by taking derivative $\frac{\partial J_2}{\partial \mathbf{w}} = 0$, thus obtained $\mathbf{w}^*$ is global optimal solution, $J_2(\mathbf{w}^{t+1}) < J_2(\mathbf{w}^t)$. □

Before proof of Lemma 6.2, we need the following **Proposition**.

**Proposition 6.3.** $\mathbf{w}^T \mathbf{F} \mathbf{w} = \sum_{g=1}^{G}(\|\mathbf{w}_{\mathcal{G}_g}\|_1)^2$.

**Proof of Lemma 6.2** Let $\Delta =$ LHS -RHS of Eq.(13). We have $\Delta$, where

$$\Delta = \sum_g \|\mathbf{w}_{\mathcal{G}_g}^{t+1}\|_1^2 - \sum_{i,g} \frac{(\mathbb{I}_{\mathcal{G}_g})_i \|\mathbf{w}_{\mathcal{G}_g}^t\|_1}{|w_i^t|}(\mathbf{w}_i^{t+1})^2 + \sum_{i,g} \frac{(\mathbb{I}_{\mathcal{G}_g})_i \|\mathbf{w}_{\mathcal{G}_g}^t\|_1}{|w_i^t|}(\mathbf{w}_i^t)^2 - \sum_g \|\mathbf{w}_{\mathcal{G}_g}^t\|_1^2 \tag{14}$$

$$= \sum_g \|\mathbf{w}_{\mathcal{G}_g}^{t+1}\|_1^2 - \sum_{i,g} \frac{(\mathbb{I}_{\mathcal{G}_g})_i \|\mathbf{w}_{\mathcal{G}_g}^t\|_1}{|w_i^t|}(\mathbf{w}_i^{t+1})^2 = \sum_g \left[ (\sum_{i\in\mathcal{G}_g} |w_i^{t+1}|)^2 - (\sum_{i\in\mathcal{G}_g} |w_i^t|)(\sum_{i\in\mathcal{G}_g} \frac{(w_i^{t+1})^2}{|w_i^t|}) \right] \tag{15}$$

$$= \sum_g \left[ (\sum_{i\in\mathcal{G}_g} a_i b_i)^2 - (\sum_{i\in\mathcal{G}_g} a_i^2)(\sum_{i\in\mathcal{G}_g} b_i^2) \right] \leq 0, \tag{16}$$

where $a_i = \frac{|w_i^{t+1}|}{\sqrt{|w_i^t|}}, b_i = \sqrt{|w_i^t|}$. Due to proposition 6.3, Eq.(14) is equivalent to Eq.(15). Eq.(16) holds due to cauchy inequality [23]: for any scalar $a_i, b_i$, $(\sum_i a_i b_i)^2 \leq (\sum_i a_i^2)(\sum_i b_i^2)$. □

**Proof of Lemma 4.1** For notation simplicity, let $\Omega_{Eg}^{\mathcal{G}}(\mathbf{w}) = \sum_g \|\mathbf{w}_{\mathcal{G}_g}\|_1^2$. Let $\mathbf{w}^*$ be the optimal solution to Eq.(11), then we have

$$0 \in \mathbf{w}^* - \mathbf{u} + \beta\partial\Omega_{Eg}^{\mathcal{G}}(\mathbf{w}^*). \tag{17}$$

In order to prove that $\mathbf{w}^*$ is also the global optimal solution to Eq.(10), i.e.,

$$0 \in \mathbf{w}^* - \mathbf{a} + \alpha\text{SGN}(\mathbf{w}^*) + \beta\partial\Omega_{Eg}^{\mathcal{G}}(\mathbf{w}^*). \tag{18}$$

First, from Eq.(12), we have $0 \in \mathbf{u} - \mathbf{a} + \alpha\text{SGN}(\mathbf{u})$, and this leads to $\mathbf{u} \in \mathbf{a} - \alpha\text{SGN}(\mathbf{u})$. According to the definition of $\Omega_{Eg}^{\mathcal{G}}(\mathbf{w})$, from Eq.(11), it is easy to verify that (1) if $u_i = 0$, then $w_i = 0$; (2) if $u_i \neq 0$, then $sign(w_i) = sign(u_i)$ and $0 \leq |w_i| \leq |u_i|$. This indicates that $\text{SGN}(\mathbf{u}) \subset \text{SGN}(\mathbf{w})$, and thus

$$\mathbf{u} \in \mathbf{a} - \alpha\text{SGN}(\mathbf{w}). \tag{19}$$

Put Eqs.(17, 19) together, and this exactly recovers Eq.(18), which completes the proof.

## Footnotes

[1]when $w_i = 0$, then $F_{ii}$ is related to subgradient of $\mathbf{w}$ w.r.t to $w_i$. However, we can not set $F_{ii} = 0$, otherwise the derived algorithm cannot be guaranteed to converge. We can regularize $F_{ii} = \left( \sum_g (\mathtt{I}_{\mathcal{G}_g})_i \|\mathbf{w}_{\mathcal{G}_g}\|_1 / \sqrt{\mathbf{w}_i^2 + \epsilon} \right)$, then the derived algorithm can be proved to minimize the regularized $\sum_g \|(\mathbf{w} + \epsilon)_{\mathcal{G}_g}\|_1^2$. It is easy to see the regularized exclusive $\ell_1$ norm of $\mathbf{w}$ approximates exclusive $\ell_1$ norm of $\mathbf{w}$ when $\epsilon \to 0^+$.

[1]Note the derivation needs Cauchy inequality [23], where for any scalar $a_i, b_i$, $(\sum_g a_g b_g)^2 \leq (\sum_g a_g^2)(\sum_g b_g^2)$. Let $a_g = \|\mathbf{v}_g\|_1, b_g = \|\tilde{\mathbf{v}}_g\|_1$, then we can get the inequality.

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
