[Reviews · NeurIPS 2014]

Submitted by Assigned_Reviewer_7 The paper proposes a new penalty term that can be used in learning a regression model; it's based on the idea that features are grouped into (possibly overlapping) groups and the penalty term discourages the use of nonzero weights for several features from the same group. The paper describes how to solve the resulting optimization problem and presents an evaluation on several synthetic and realistic datasets, demonstrating that the proposed approach works better than several alternatives. The paper is clearly written and well organized, and seems to present an interesting incremental improvement on various earlier LASSO methods (which are mentioned in the related work section).
Comments for the authors:
"exclusive group LASSO" [caption of Fig. 1] -- should be "Exclusive" "Eq.(2)" [caption of Fig. 1] -- space missing; there are several similar occurrences elsewhere
"G_g \in P({1, 2, ..., p}) denotes a set of group g" [page 2] -- I found this sentence unclear. What exactly is the relationship between g and G_g? Earlier you say that a group g is a subset of {1, 2, ..., p}, and later you give an example where G_g = {1, 2, 4}, so it would appear that G_g is also a subset of {1, 2, ..., p}; what then is the difference between g and G_g at all?
"feature index u \in supp(g) will be non-zero" [page 3] -- I found this sentence a bit unclear as well. What exactly is this feature index u, how is it chosen and what is its purpose?
On page 3, there are several instances where groups are described using square brackets: G_u = [1, 2], G_v = [2, 3] and so on. Why do you use brackets here whereas on page 2 you used curly braces (e.g. "G_g = {1, 2, 4}")? This makes it seem as if the groups were now real-number intervals, but probably that's not intended here.
"is a diagonal matrices" [page 4] -- should be "matrix"
I was slightly surprised when I saw that your experiments on realistic data all focus on using your improved LASSO (and other methods) for feature selection (and then applying SVM to train a classifier); when I read about the definition of LASSO in [21], I was under the impression that its original purpose was to improve the training of regression models, and your eq. (2) seemed to be along the same lines as well, so I thought that you would use some regression tasks to evaluate the method in your experiments. It might be useful to add some discussion on why you use the method only for feature selection (in the context of a classification task), and not for any regression tasks. Additionally, it might be good to add some of the more traditional feature-selection methods for comparison (e.g. information gain etc.).
Above the first table on p. 7, you say "we list the mean and variance of classification accuracy", but it wasn't quite clear to me what the mean and variance are taken over. Did you perform some sort of cross-validation or something like that?
Summary: I think this is an interesting paper, a nice incremental improvement on existing work, it seems technically sound and presents a reasonable experimental evaluation. Submitted by Assigned_Reviewer_13 The authors present a novel Lasso regressor formulation called “exclusive group LASSO” that is supposed to enforce the sparsity for features at an intra-group level. The method is to allow sparsity at intra-group level in the context of feature selection and not only inter-group like SoA group-LASSO. In this new formulation, In intra-group sparsity is obtained through a ℓ1 penalization, while inter-group non-sparsity with a ℓ2 penalization term. A second aspect of the paper is the proposition of a proved convex formulation of the associated Loss function. Finally, SoA synthetic and UCI dataset are used to illustrate the capability of the method. Quality : Sufficiently well written and illustrated. Clarity : The paper is quite readable and the propositions are formally justified. More challenging dataset would have been prefered than synthetic one perhaps. Originality and Signifiance : despite the large amount of already existed and studied LASSO algorithm, the exclusive group LASSO model and associated solving technique proposed in this paper is novel and justified enough to be signifiant.
Summary: An interesting original formulation of a well known and studied approach of feature selection, group LASSO. The approach is well presented, justified and theoretically grounded. More challenging experiments would have been prefered. Submitted by Assigned_Reviewer_24 This paper presents a new formulation known as Exclusive Group LASSO for enforcing the sparsity for features in Group LASSO at an intra-group level. Both theoretical convergence analysis and empirical study are given. Summary: The intra-group sparsity are shown to outperform traditional LASSO in several experiments, but this result may be dependent on how closely related these feature groups are. Thus a cauterization of the feature relations may be important to consider. Author rebuttal: We thank all reviewers for their valuable comments and strong support.
To reviewer 1:
Thanks for your support on our paper. Your main concern is to include some experiments and comparisons on more ``challenging'' dataset. Due to space limitation, we only report a portion of our results. We will include more experiments (such as ANDI dataset, pie face image dataset) in the supplemental materials of our final version.
To reviewer 2:
Thanks for your strong support and insightful comment. We agree that the results depend on how closely related these feature groups, which has been partially indicated by the correlation graph in Fig. 2c. It is kind of an implicit assumption behind of our formulation (exclusive group LASSO). Thanks for you pointing out this. We will include more examples / graphs to illustrate this point clearer.
To reviewer 3:
We very appreciate your carefully reading. We have fixed typos you pointed out. A few confusions are clarified in the following: G_g is a subset of {1, 2, ..., p}, g=1,2,3, ..., G; G is the total number of the group. G_u = {1, 2}, G_v = {2, 3}, and so on. Square brackets should be brackets. From the definition of supp(g) in Page 2, any feature index u that falls in set supp(g) is corresponding to non-zero feature value.
We also tried experiment on regression tasks. The experiment results also suggest the advantage of the exclusive group lasso also over several other lasso type formulations. Due to space limitation, they were not shown in our original submission, but would be included in our final version (probably in the supplemental material). In addition, comparisons to some traditional feature selection methods (like information gain) would be added in our final version.
We use different methods to select features, and then feed the selected features to linear SVM. In classification task, we use five-fold cross validation, and list the mean and variance of classification accuracy in Page 7.
| |